# Occult Neck Metastases in Head and Neck Adenoid Cystic Carcinoma: A Systematic Review and Meta-Analysis

**DOI:** 10.3390/jcm11164924

**Published:** 2022-08-22

**Authors:** Jacopo Zocchi, Matteo Campa, Giulia Bianchi, Oreste Iocca, Pasquale Di Maio, Gerardo Petruzzi, Silvia Moretto, Flaminia Campo, Armando De Virgilio, Vincent Vander Poorten, Raul Pellini

**Affiliations:** 1Department of Otolaryngology Head and Neck Surgery, Regina Elena National Cancer Institute IRCCS, 00144 Rome, Italy; 2Division of Otolaryngology and Head and Neck Surgery, European Institute of Oncology IRCCS, 20142 Milan, Italy; 3Otolaryngology, Clinical and Sciences Translational Medicine Department, University of Rome “Tor Vergata”, 00144 Rome, Italy; 4Ear-Nose-Throat & Audiology Unit, University of Ferrara, 44121 Ferrara, Italy; 5Division of Maxillofacial Surgery, Città Della Salute e Della Scienza Hospital, University of Torino, 10125 Torino, Italy; 6Department of Otolaryngology-Head and Neck Surgery, Giovanni Borea Civil Hospital, 18038 Sanremo, Italy; 7Department of Biomedical Sciences, Humanitas University, Via Rita Levi Montalcini 4, 20090 Milan, Italy; 8Otorhinolaryngology Unit, IRCCS Humanitas Research Hospital, Via Manzoni 56, 20089 Milan, Italy; 9Otorhinolaryngology-Head and Neck Surgery and Department of Oncology, Section Head and Neck Oncology, Leuven Cancer Institute, University Hospitals Leuven, 3000 Leuven, Belgium

**Keywords:** cN0 neck, adenoid cystic carcinoma, cervical occult lymph node metastases, elective neck dissection, meta-analysis, salivary gland

## Abstract

**Introduction:** Adenoid cystic carcinoma (AdCC) is a rare tumor whose clinical course is burdened by local recurrence and distant dissemination. Lymph node metastasis is not believed to be common and its clinical impact is controversial. The aim of this study was to determine: (1) the prevalence of occult metastasis at diagnosis in cN0 head and neck AdCC, (2) its prognostic role, and (3) the consequent need to perform elective neck dissection (END). **Material and Methods:** A systematic review and meta-analyses following PRISMA guidelines was performed. PubMed, Embase, and Central databases were questioned up to July 2021 to identify studies reporting on the prevalence of occult neck metastases in head and neck AdCC. A single-arm meta-analysis was then performed to determine the pooled prevalence of occult lymph node metastases among the retained studies. **Results:** Of the initial 6317 studies identified, 16 fulfilled the inclusion criteria, and they were included in the meta-analysis. Of a population of 7534 patients, 2530 cN0 patients were treated with END, which revealed 290/2530 cases of occult metastases (pN+/cN0). Meta-analysis of the results of END in the 16 studies estimated an overall prevalence of occult metastases at diagnosis of 17%. No further subgroup analysis was possible to identify factors influencing lymph node involvement and the prognostic role of END. **Conclusions:** Taking 20% as an historically proposed cut off, a 17% prevalence of occult metastases represents a borderline percentage to get a definitive conclusion about the indication to END for head and neck AdCC. A more advanced UICC stage, an oropharyngeal minor salivary glands origin, and a high-grade transformation are factors to be considered in a comprehensive patient’s tailored therapeutic strategy. Multicenter prospective studies are the key to finding stronger recommendations on this topic.

## 1. Introduction

Adenoid cystic carcinoma (AdCC) is a rare tumor; it accounts for 1% of all head and neck malignancies [1,2], but it is one of the most frequent histotypes of the salivary gland carcinomas. It has a relentless growth pattern associated with local recurrences and late distant hematogenous dissemination to the lung, liver, bone, and brain. On the other hand, lymphatic spread has long been considered not a typical figure of this disease and, differently from squamous cell carcinoma (SCC), prevalence of occult lymph node metastasis is less likely and the association between the presence of occult lymph node metastases and survival remains inconclusive. Consequently, the optimal neck management for patients affected by AdCC has been the subject of many studies in the recent literature and no definitive recommendations have been reached [3,4]. As surgical or radiotherapeutic neck treatment is not free from morbidity, the decision regarding whether to treat or not should be based on the estimated prevalence of occult lymph node metastases and on the expected impact of their treatment on survival. The aim of this study was to determine the prevalence of occult metastases in an attempt to establish the possible clinical impact of elective neck treatment for patients affected by head and neck AdCC.

## 2. Methods

This systematic review (SR) and meta-analysis (MA) followed the guidelines of the Preferred Reporting Items for Systematic Reviews and Meta-Analyses (PRISMA).

A systematic search of all articles published before July 2021 was performed on PubMed, Embase, and Central. Combinations of MeSH terms and free-text words were utilized to search for “Adenoid Cystic Carcinomas” AND “Salivary Gland” AND “Neck dissection” OR “Surgery”. For these keywords, all synonyms were used.

### 2.1. Inclusion Criteria

Two independent reviewers (J.Z. and M.C.) screened all papers on the title and the abstract, with the following inclusion criteria:-Patients affected by head and neck salivary gland cancers, previously untreated and with surgery as primary therapy;-Studies with complete or extractable data on the number of patients with negative neck at presentation (cN0), number of elective neck dissections performed, and pathological finding of metastases (pN0).

Excluded were papers reporting on recurrent cases, animal, cadaveric, and radiological studies; as well as unobtainable full-text studies, irrelevant studies, and studies with insufficient or aggregated data; non-original studies (i.e., reviews, editorials, and letters); papers not in English; and studies reporting on less than 10 patients.

We contacted the authors of the selected studies in order to collect missing data about an individual patient and to attempt to perform subgroup meta-analysis.

Concordance between the two reviewers was calculated with a Cohen’s k test, with a result of k = 0.71. Any disagreement between the reviewers on the eligibility of articles for inclusion was settled by discussion or, failing this, by a referral to a third author (G.B.).

### 2.2. Data Extraction and Statistical Analysis

The following data were extracted from each of the included studies: author, year of publication, study design, country and period of conduction, number of patients, demographic characteristics, UICC TNM stage, histology and grading of AdCC, criteria adopted for END and type of neck dissection, number of clinically negative patients who underwent END, and cases of occult metastases identified (pN+/cN0). Occult metastases was defined as the presence of nodal metastases upon END of patients with a previous clinically negative cervical lymph node.

A single-arm MA of the rate of occult metastases in cN0 patients undergoing END was performed using the R software for statistical computing (R 2.10.1; “meta” package); arcsine proportion transformation of the data was performed, and the restricted maximum likelihood method was applied for random effects meta-analysis. A 99% confidence interval (CI) was set for the analysis, which is desirable, given the observational nature of the included studies, and it should lead to more conservative results. The heterogeneity among the included studies was evaluated by I2 statistic, and the Cochrane criteria were taken as a reference to estimate the level of heterogeneity.

## 3. Results

### 3.1. Study Selection

The bibliographic research led to the initial identification of 6317 studies. Details on the literature search process are shown in the flow chart of Figure 1. By reading the titles and abstracts, 155 full-text articles that reported on primary surgical treatment of untreated head and neck AdCC were analyzed. Of these, 138 were excluded as they did not meet the inclusion criteria. Finally, 16 articles fully satisfied the inclusion criteria [5,6,7,8,9,10,11,12,13,14,15,16,17,18,19,20]. Reasons for study exclusions are detailed in the flow diagram (Figure 1), with a comprehensive population of 7497 patients.

### 3.2. Included Studies

The 16 articles that satisfied the criteria for inclusion in the SR and MA were all retrospective case series published between 1995 and 2020. Patient demographics were available in 15 studies. The 7481 patients included in these comprised 3173 (42.4%) males and 4308 (57.6%) females, whilst the mean age was 55.39 years (4.53 SD, 18–91 range years). Tumor location was extractable from all the studies, and it was distributed as follows: 4460 (59.5%) in the major salivary glands and 2885 (38.5%) in the minor salivary glands; the site was unknown in 152 patients (2%). In particular, AdCC was located in the parotid gland in 1416 (18.79%) patients, 129 (1.71%) in the sublingual gland in, and 1220 (16.19%) in the submandibular gland. Oral cavity minor salivary glands were the primary site in 1362 (18.07%) patients, sinonasal in 644 (8.54%), and “other site” in 2763 patients (36.67%). The tumor grade was available for 427 (5.66%) patients, with low and intermediate-grade AdCC being reported in 353 (4.68%) and high-grade AdCC in 74 (0.98%). Of the 7087 (94.5%) patients in which T-classification was reported, 4020 (56.7.%) were T1/T2 and 3067 (43.3%) T3/T4. The demographic and the histological features of the included studies are summarized in Table 1.

### 3.3. Meta-Analysis

The management of the neck was an END in 2556 cN0 patients of the 7534 with head and neck salivary gland AdCC, which led to the diagnosis of 294/2556 cases of occult metastases (pN+/cN0). The MA of the results of END in the 16 studies estimated an overall prevalence of occult metastases of 17% (99% CI 10-25.3) (Figure 2). The I2 was 88.4%, indicating that the heterogeneity between the 16 studies included in the statistical analysis was high.

Indications for END were reported in 7/16 studies: most of these report “surgeon’s preference” or “institution’s protocol” as the main factor to carry out END [6,7,11,19]; more detailed factors were: advanced T-classification of tumors and patients where the neck had to be opened as part of a transcervical approach for access to the tumor or for reconstruction of the primary defect using a free flap [5]; invasion into the bone tumor location in the retromolar trigone or lip [18]; patients with high-grade malignancy or dedifferentiated AdCC [14]. Only five studies specified the extent of prophylactic surgical neck treatment, with selective dissection of levels I-III as the most reported [6,11].

Only two studies performed a subgroup analysis based on the tumor site [6,20], and only one [20] upon T classification at presentation, thus preventing a possible subgroup meta-analysis.

Only one study analyzed the possible relationship between pathological adverse features (high-grade histology or perineural invasion) and occult metastases, without finding a significant correlation [6].

## 4. Discussion

Historically, a 20% rate of occult neck metastases has been proposed as a sensible cut off of whether to perform an END or not for SCC, balancing gain in oncological control with possible treatment-related complications. Concerning AdCC, occult lymph node metastases have not been confirmed as a definitive factor impacting the course of the disease. Clinically evident metastases at diagnosis have been associated with a worse outcome and a prevalence between 19% and 37% has been found, depending on the series and site of origin of the tumor [21,22]. On the other side, the prevalence of occult neck metastases has been scarcely reported in the literature.

Despite this uncertainty, neck dissection is frequently performed in association with primary tumor removal. In a series of definitively treated AdCC by Cassidy et al. based on the National Cancer Database, 70.4% of the population studied (2209 patients) underwent neck dissection [9]. In another work based on the same database, END was mostly performed in academic centers compared to community facilities and also for patients that had been transferred between facilities for treatment [20]. These data show that END is frequently performed when dealing with head and neck AdCC, especially by head and neck surgeons at more specialized centers that include END as part of the treatment regimen [6].

In this setting, more evidence to justify a surgical procedure that is not free of morbidity is desirable. Our analysis focuses upon AdCC as a whole. The material did not allow subgroup-analysis by site, UICC staging, or histopathological features. Nevertheless, AdCC across sites and stages is marked by a constant relentless aggressivity, with differing speed of development according to the grade of the disease. Indeed, most of the series do not study the association of clinical or pathological factors with the presence of occult neck metastases, preventing a more in-depth subgroup meta-analysis.

Our systematic review concludes that in a population of 2530 AdCC patients who underwent END, the pooled prevalence of occult metastases was 17%. This is a not neglectable prevalence but a borderline result for justifying an elective procedure per se. Going beyond a pure prevalence, other factors have to be taken into consideration in order to interpret this overall number of 17% and link it to the decision on whether to proceed to END.

## 5. Site of Disease

AdCC can originate from major and minor head and neck salivary glands. Different tumor origin seems associated with a different probability for lymphatic dissemination, intrinsically influencing the necessity of an END, because of a different lymphatic density and/or permeability. In their study upon incidence of cervical lymph node metastasis, Amit et al. report an overall rate (cN+ pN+) of neck metastases of 29%. The rate observed in the oral cavity (37%) was significantly higher than that of major salivary glands, 19% (*p* = 0.001) [21]. In the population of our study, the major salivary gland was the most represented tumor site (58% of case), while the oral cavity was the most frequent between minor salivary glands (18%). Only two studies in our MA correlated site and occurrence of occult neck metastases in their analysis: one reported that the major salivary glands and tongue are most distinctive with regard to the frequency of both END and subsequent identification of occult nodal metastases. In this study, END for tongue AdCC repeatedly uncovered occult nodal metastasis more than 20% of the time, regardless of the staging. In contrast, patients with AdCC of the nasal cavity/nasopharynx (12.0%, 3 of 25), hard/soft palate (9.5%, 2 of 21), oral cavity (15.4%, 4 of 26), floor of mouth (11.1%, 2 of 18), and larynx (15.4%, 2 of 13) did not exceed this threshold [20].

The second study reported an overall prevalence of occult nodal metastases among the patients who underwent END of 17% (38/226), a result that is higher among patients with AdCC of the oral cavity (22%, 25/116 patients), lower among those with sinonasal AdCC (17%, 4/24), and the lowest among patients with a major salivary gland AdCC (11%, 9/85 patients). This difference was not statistically analyzed, but it was noticed that the decision to perform the elective neck dissection was significantly influenced by the tumor site [6]. Neck metastases were retrieved in 4 of 9 patients (44%) in a series of patients affected by oropharyngeal and oral cavity minor salivary glands AdCC [13] and only in 2 of 19 (10%) patients with submandibular gland AdCC [10]. In the study by Mücke et al., none of the ten patients with minor salivary glands AdCC who underwent END presented with occult neck metastases, but 8 of 33 presented (24%) with a clinically positive neck node [18]. Looking at the literature, the submandibular gland is slightly burdened by node metastasis (22.5%) compared to the parotid gland (14.5%) [4]; oral cavity and oropharyngeal minor salivary glands are confirmed to have a major tendency for node metastasis, with the series reporting up to 43% of cases [22]. This behavior could be explained by their advanced stage at presentation and the extensive lymphatic network in this site [20,22,23]. Sinonasal AdCC seems to manifest a minor tendency for neck node metastases both at presentation [24,25] and long-term distance [26]. A collective international review estimated a prevalence of 12.1% nodal metastases in a population of 91 laryngeal AdCC patients [27].

The only study to compare the impact of END on outcome according to the primary tumor site (oral cavity, major salivary glands, or sinonasal salivary glands) showed survival rates similar to those for the patients with and without END [6].

### T-Classification

Local extension of the primary lesion should logically predict occult neck metastases occurrence. The only study to perform this kind of analysis reports that the odds of performing END paralleled increasing clinical T classification. Compared to T1 patients, T2 (OR 1.57, 95% CI 1.22–2.02), T3 (OR 2.17, 95% CI 1.61–2.91), and T4 (OR 3.02, 95% CI 2.24–4.08) were all associated with an increased rate of END (*p* < 0.001). Using logistic regression, clinical T-classification significantly predicted occult nodal metastasis among END patients. If a higher stage has been reported as a criterion to perform END [18], quite surprisingly Bhayani et al. report that of 30 early-stage AdCC who underwent elective neck dissection, 7 had occult neck metastases (23%) and 6 had extracapsular spread (ECS), encouraging END also for this population [8].

Again, Xiao highlights that END did not lead to an advantage in OS when dealing with T1–T2 lesions, while it did for T3-T4 lesions (5-year OS 78.1% vs. 70.4%, *p* = 0.041).

## 6. Prognostic Role of Occult Metastases

Between the studies selected in our meta-analysis, seven questioned the association between lymph node involvement and survival: three found an association with a worse survival [6,15,18], whereas four studies did not [11,13,14,17]. Lee et al., studying 61 patients with head and neck AdCC, found an overall survival rate of 85% at 5 years and 81.1% at 10 years in patients with an overall negative N-status [15] against 56.8% at 5 years and 28.4% at 10 years in patients with an overall positive N-status. On the other hand, there was no significant difference for OS between pN0 and pN+ (*p* = 0.366) in the study by Cordesmeyer [11]. In particular, these authors found a DFS of 85.7% at 5 and 10 years for the group with pN+, whereas the amount was 58.6% after 5 and even at 10 years in the group with pN0. Despite the findings in this one study, pathological lymph node involvement with or without extracapsular spread at diagnosis in AdCC is recognized as an independent prognostic factor in most reports [28,29,30]. Lymph node involvement was the only factor associated with decreased OS on multivariate analysis in an American study of 110 patients [31] and in a Japanese study of 42 patients [32].

Moreover, most authors report lymph node involvement as a risk factor for subsequent occurrence of distant metastasis. In the study of Bhayany et al., ECS and solid tumor subtypes were independently associated with the development of DM [8]. This is confirmed by the work of Kawakita et al. that, even without noting an association between pN+ and survival, demonstrates an impact on distant metastasis free survival [14]. Once more, a 5-year distant metastasis rate was significantly higher among patients with pN+ than among those without (40% and 27%, respectively) in the work by Amit et al. [6]. According to Ko et al., 75% of patients with an initial nodal involvement eventually developed distant metastasis [32].

## 7. Therapeutic Role of END

Besides the prevalence of occult metastases, more importantly, the effectiveness of a treatment should rely on a proven oncological advantage. Unfortunately, there remains uncertainty about the impact of END on locoregional control (LRC), disease free survival (DFS), and overall survival (OS), due to conflicting results reported in the current literature.

Among the studies included in our MA, only one reports a benefit from an END as compared to three that do not see an advantage (see Table 2). As already noted, Xiao reports that patients with advanced-stage disease who underwent surgery alone experienced a significantly inferior OS compared to those who underwent surgery with END (5-year OS 78.1% vs. 70.4%, *p* = 0.041) [20]. On the other hand, Amit reports a 5-year OS of 72% for the patients who underwent END, compared with 79% for the patients who did not (not statistically different) and also the 5-year regional control and distant metastasis rates did not differ significantly between the two groups [6]. The same result was found by Cordesmeyer et al. [11]. Even if they found a non-statistically significant higher 15-year survival rate in the END group (86% for END and 64% for group without END, *p* = 0.829), DFS between those two groups did not differ significantly (65.6% at 5 and 10 years for the END group and 81.0% and 26.7% for the group without END, respectively). In the study by Kawawita, among cN0 cases (*n* = 161), neck dissection did not improve OS or LRC [14]. The reliability of these results suffers from a likely selection bias: patients undergoing END are likely those with more advanced disease and worse prognostic factors that are incorporated in the decision to perform more extended surgery.

## 8. Elective Neck Irradiation

Neck disease control could take advantage of postoperative elective neck irradiation (ENI). The use and the indication for radiotherapy, however, does not seem to be homogeneous in the current literature.

In the studies selected in our MA, only Balamucky et al. analyzed the role of ENI [7]: of 102 patients presenting with an undissected clinically negative neck (cN0), 38 patients (37%) were observed and 64 patients (63%) received ENI. A better disease control was observed for the ENI arm, controlling for confounders in multivariate analysis (10y neck control 95% with ENI vs. 89% without ENI, *p =* 0.0469). The authors do not specify which factors led to the use of RT, certainly patients were not randomly assigned to the two groups. Moreover, of 11 patients who received END, 9 had pathologically negative neck nodes, yet 6 of these did receive postoperative ENI, but the reasons for this choice are not reported. Of the remaining three patients, one developed a recurrent disease in the neck and lungs.

Despite a lack of standardization, adjuvant therapeutic lateral neck irradiation seems to be widely adopted: in the study of Amit, 66% of patients who underwent END, compared with 55% of patients without END (*p* = 0.09) received postoperative radiotherapy, with a dose ranging from 60 to 74 Gy [6]. A major tendency for adjuvant neck irradiation for patients who underwent END is confirmed by Xiao et al. [20]. A total of 14 of 34 (41%) patients received adjuvant radiotherapy in the study by Cordesmeyer et al. [11] and 21 of 50 (42%) in that of Lee et al. [15].

Shen reports that 25 patients received ENI at the discretion of their attending radiation oncologists, mainly in patients with extensive infiltration of the primary disease [16] as well as in the study by Agarwall in which ENI was offered selectively to the patients with primary sites rich in capillary lymphatics [33].

Garden et al. reasonably report to include the neck in the irradiation field in case of pathological nodes [12], and they also rightly state that the upper neck nodes will often be included in the field required to cover the primary tumor. In this study, none of the 20 irradiated patients with pathologic nodal disease recurred in the irradiated neck area. 

## 9. Limitations

Despite adherence to PRISMA guidelines and adoption of strict inclusion and exclusion criteria, some limitations of this MA have to be highlighted: the most important is the retrospective nature of all selected studies and thus an unavoidable selection bias; and, secondly, the impossibility to perform any subgroup analysis according to site, staging, and treatment. Furthermore, criteria for selecting patients in many studies is either insufficiently stringent or inadequately described.

Finally, our SR and MA were not able to determine which therapeutical strategy (i.e., END, ENI, or observation) gives the better oncological outcome and therefore its prognostic role.

## 10. Conclusions

By pooling the available information in the literature on head and neck AdCC, this systematic review and meta-analysis offers a solid estimate of an overall prevalence of 17% occult regional nodal metastasis. While the correlation between clinically obvious neck metastases and survival and distant metastases is recognized, the oncological impact of treating occult neck metastases, however, remains controversial.

Until randomized prospective studies bring unbiased evidence on this topic, the indication for END remains patient-tailored. More advanced UICC-stage disease presentation, the oropharyngeal minor salivary glands as a site of origin, and high-grade transformed AdCC are probably the entities that will benefit from performing this procedure. Being in favor of END leads to the observation that it does not add significant morbidity when dealing with more extended lesions that necessitate an open neck procedure or a flap inset, and it allows for an exact pathological N status, which is important for the decision on the extent of further treatment, such as radiotherapy in the case of positive nodes. On the other hand, keeping the unproven oncological impact in consideration, refraining from END in early-stage disease and frail patients is a defendable approach. In addition, ENI can be a reliable strategy if an adjuvant radiotherapy is already foreseen for reasons related to the primary tumor or when it becomes necessary due to the pathological features of the resected primary.

Once again, we have to underline that none of the studies in the literature specify the indications for selecting patients both for END and ENI, so selection bias limits the value of these reports.

## Figures and Tables

**Figure 1 jcm-11-04924-f001:**
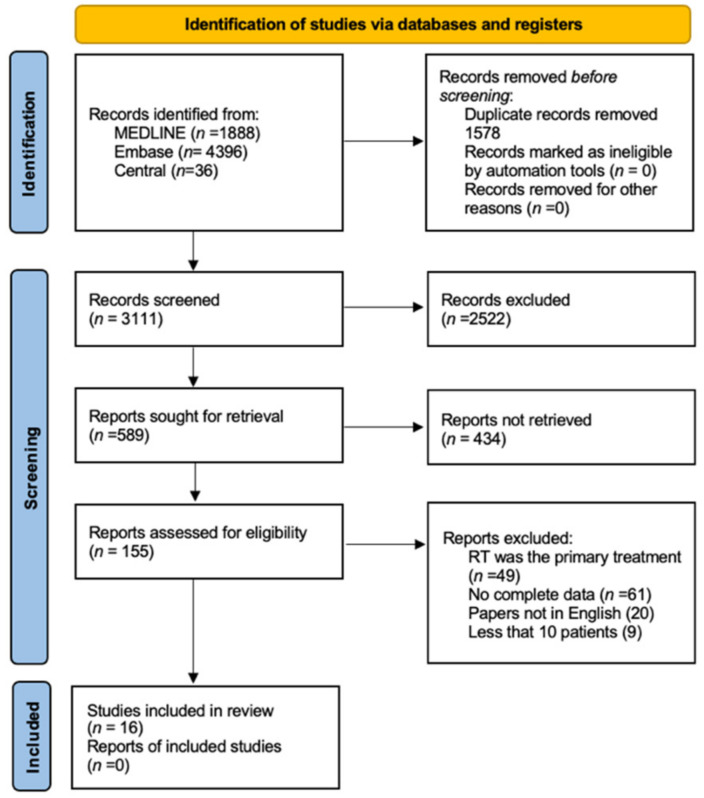
Flow chart of study inclusion process. RT: radiotherapy.

**Figure 2 jcm-11-04924-f002:**
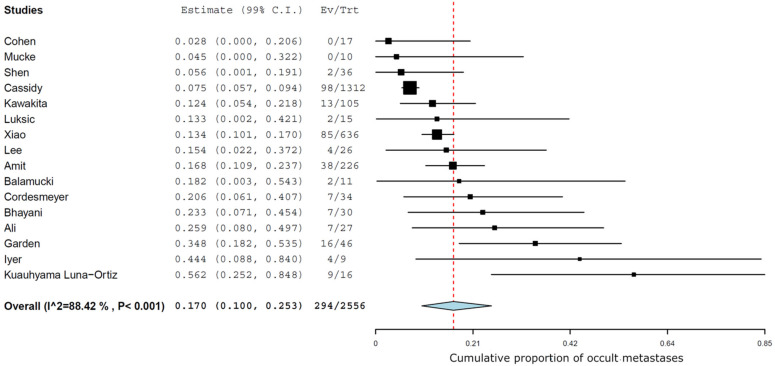
Forest plots showing results of elective neck dissection in the included studies [5,6,7,8,9,10,11,12,13,14,15,16,17,18,19,20]. Ev: events; Trt: treatments.

**Table 1 jcm-11-04924-t001:** Demographic and clinical features of included studies.

Demographic and Clinical Features of Included Studies											
Study	N° Pts			Median Age (yrs.)	Primary Site			Stage			Median FU (mhs)
	Tot	M	F		Minor Salivary Glands	Major Salivary Glands	Unknown Site	T1–T2 (%)	T3–T4 (%)	Stage NA	
Ali, 2017 [5]	87	25	62	54	59	28	0	58 (66.7%)	29 (33.3%)		NA
Amit, 2015 [6]	457	190	267	56	324	133	0	288 (63%)	169 (37%)		64
Balamucki, 2012 [7]	120	60	60	56	96	24	0	51 (42.5%)	68 (56.7%)	1 (0.8%)	103.2
Bhayani, 2012 [8]	60	23	37	52	0	60	0	60 (100%)			72.5
Cassidy,2019 [9]	3136	1286	1850	55.8	571	2416	149	1539 (49.1%)	1531 (48.8%)	66 (2.1%)	58.4
Cohen, 2004 [10]	22	7	15	48	0	22	0	13 (59%)	9 (41%)		67
Cordesmeyer, 2018 [11]	61	27	34	56.4	46	15	0	0	0	61 (100%)	71.2
Garden, 1995 [12]	198	87	111	50	127	71	0	0	0	198 (100%)	NA
Iyer, 2010 [13]	16	NA	NA	NA	16	0	0	13 (81.25%)	3 (18.75%)		86
Kawakita, 2020 [14]	192	128	64	61.5	43	146	3	89 (46.4%)	54 (28.1%)	49 (25.5%)	66
Lee, 2014 [15]	61	23	38	53.37	31	30	0	40 (65.6%)	21 (34.4%)		58
Lukšić, 2016 [16]	45	21	24	53.4	29	16	0	30 (67%)	15 (33%)		129.4
Luna Ortiz, 2016 [17]	101	32	69	50	78	23	0	30 (29.7%)	36 (35.6%)	35 (34.6%)	52
Mucke, 2010 [18]	33	18	15	63.69	33	0	0	21 (63.6%)	12 (36.4%)		112.2
Shen, 2012 [19]	101	65	55	NA	47	54		60 (59.4%)	41 (40.6%)		65
Xiao, 2019 [20]	2807	1200	1607	57.7	1385	1422	0	1728 (61.5%)	1079 (38.50%)		NA

NA: not available; FU: follow up; Tot: total; M: male; F: female.

**Table 2 jcm-11-04924-t002:** Summary of criteria adopted, type, and number of neck dissections performed.

Neck Dissection									
	N° Tot	N° Pts Underwent ND	Criteria For ND	Type of ND Performed	cN		pN		Comments
					N0	N+	N0	N+	
Shen, 2012 [19]	101	40	Clinically positive nodes Surgeon preference	NA	97	4	95	6	- Adj radiotherapy significantly improved locoregional control and DFS rates for ACC as compared to surgery alone - Increasing tumor stage was an adverse determinant for treatment outcomes - Surgical margin status or perineural invasion were not significant
Amit et al., 2015 [6]	457	226	Institutional protocol of treatment	END, MRND	457	0	179	47	- Incidence of occult neck metastases among pts with ACC was 17% - No survival advantage for pts who underwent END compared with those who did not, regardless of site
Ali et al., 2017 [5]	87	30	Clinically positive nodes; Large T-stage tumors; Cases where the neck had to be opened as part of a transcervical access Any case where free flaps were indicated for reconstruction of the primary defect	NA	84	3	20	10	- Local control rate of 89% at 5 years and 79% at 10 years - Pathological T3/T4 status and no PORT are independent predictors of local failure - PNI and negative neck nodes do not receive PORT - Pts who did not receive PORT had a significantly poorer local control rate - Pts who did not receive PORT were 13 times more likely to fail locally
Lukšić et al., 2016 [16]	45	15	No strict criteria	RND, SND	45	0	13	2	- The rate of occult neck disease was 26.7% - Impact of neck dissection on survival was beyond the scope of this study and not analyzed
Mucke et al.,2010 [18]	33	18	Suspicious enlarged lymph nodes by ultrasound sonograph, in the CT or MRI > 1 cm Central necrosis found in the loco-regional lymph nodes, Invasion into the bone Patients with tumor location in the retromolar trigone or lip underwent neck dissection	NA	25	8	10	8	- Neck dissection considered if lymph node involvement is possible or T category is at higher stage - Neck can be treated prophylactically in N0 patients with lesions of the lower part of the oral cavity
AN Cohen et al., 2004 [10]	22	19	NA	SND, MRND, RND	20	2	17	2	- Neck dissections should be performed only in patients with known local lymph node involvement
K Luna Ortiz et al., 2016 [17]	101	16	NA	NA	90	11	7	9	- Only 5.6% of our cases had positive adenopathy at surgery - N does not seem to clearly influence on survival
Kawakita et al., 2020 [14]	192	121	Patients with lymph node metastasis And/or high-grade malignancy indicated preoperatively by aspiration cytology	NA	161	16	144	29	- Neck dissection did not improve clinical outcomes in patients with cN0
Cordesmeyer et al., 2018 [11]	61	34	Surgeon preference	END	61	0	0	7	- No significant difference in the OS and the DFS in patients with pN+ vs. pN0 - END is recommended to lower the chances of loco-regional recurrence - The exact pathological N status might be important for further cellular analysis or a tighter follow up treatment
Xiao et al., 2019 [20]	2807	636	NA	END	2807	0	551	85	- END showed significantly extended OS for the subset of patients with advanced ACC of the major salivary glands (T3–T4) compared to patients who underwent resection alone - Combination of surgery with END and adjuvant XRT predicted significantly extended OS
Cassidy et al., 2019 [9]	3136	2327	NA	END	2059	194	1855	354	- Rate of unexpected nodal disease after elective neck dissection was 7.5%
Bhayani et al., 2012 [8]	60	30	NA	END, MRND	60	0	23	7	- Proportion of patients developing DM was significantly greater in patients with N+ after elective neck dissection
Iyer et al., 2010 [13]	67	39	NA	END, Therapeutic ND	47	20	17	22	- 25% cN0 necks in ACC harbored occult metastasis, suggesting ipsilateral END
Balamucki et al., 2012 [7]	120	11	Surgeon preference	END	113	7	9	2	- It is prudent to electively treat the first group nodes especially in patients with primary tumors in sites rich in capillary lymphatics
Garden et al., 1995 [12]	198	50	NA	NA	NA	NA	30	20	- Treat the neck only when nodes are involved, although by the nature of the field required to cover the primary tumor, the upper neck nodes will often be included
Lee et al., 2014 [15]	61	30	NA	END, MRND	57	4	22	8	- No significant differences in distant metastasis or survival between END and no END groups

ND: Neck Dissection; SND: selective neck dissection; MRND: modified radical neck dissection; RND: radical neck dissection; OS: overall survival; DSS; disease specific survival; LCR: local control rate; DMFS: distant metastases free survival; CT: computed tomography; MRI: magnetic resonance imaging.

## Data Availability

The data that support the findings of this study are available from the corresponding author upon reasonable request.

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
