# Peer review of "Occult Neck Metastases in Head and Neck Adenoid Cystic Carcinoma: A Systematic Review and Meta-Analysis"

_jcm, 2022, doi:10.3390/jcm11164924_

Round 1

Reviewer 1 Report

Dear Authors, Thank you for the submission of your manuscript "occult metastases in head and neck adenoid cystic carcinoma: a systematic review and meta-analysis."

These are my comments

*For your paper to provide useful information, it is suggested to set a number of "key questions" at the beginning of the paper as a framework for your analyses.

For example, Does tumour site (parotid vs. submandibular vs. oral cavity vs. other) influence the incidence or neck levels of nodal metastases?

Does T stage influence risk of metastases?  ie T1/2 vs. T3/4

Does histology  low/intermediate vs. high grade influence risk of metastases?

For similar stages, did the absence / presence of neck dissection influence prognosis?  ie for T3/4 tumours that had neck dissection but were pathologically negative - did they fare the same as T3/4 tumours that did not get neck dissection?  DFS? OS?

In order to obtain these data, you may need to go back to the original papers to see if these specific data are available - In some cases, you may need to contact the original authors to see if you can get access to the anonymized data set containing individual data from their manuscripts.

Only in this more detailed analysis and using statistical methods can your manuscript provide useful information.

Otherwise a crude rate of "17%" rate of nodal metastases for the entire cohort is not a very meaningful statement. 

Regards

Author Response

We appreciate reviewer’s observation and we strongly agree with him.

Aim of this study was to establish the rate of occult neck node metastasis in the adenoid cystic carcinoma and, more interestingly, factors influencing this rate. In all the papers selected we retrieved data about site, staging, grading or prognostic role of END. We contacted all the authors in order to get more data about our search. Unluckily the paucity of data retrieved prevented any subgroup analysis. This is explained both in the results and discussion section, where we highlighted the limitations of our analysis. We tried to overcome this limit trough a descriptive report of the results of each paper selected, presenting dedicated sections in our discussion. We can agree that a crude rate of nodal metastases is not very meaningful statement. On the other side, this is a rigorous statistical analysis that followed PRISMA guidelines and bring a solid result about a debated topic, highlighting the limits of the studies of the contemporary literature and thus pushing on more meticulous future reports.

Reviewer 2 Report

Zocchi and collegues present an interesting review and meta analysis regarding the occult metastases in head and neck adenoid cystic carcinoma. 

The meta analysis is correctly performed and the review of literature is exhaustive.  Unfortunately, it was not possibile to perform any subgroup analysis according to site and staging.

The Authors made an exhaustive review of the literature about the neck metastasis in head and neck  metastases, they carefully discussed the main points (site of disease, prognostic role of occult metastases, therapeutic role of elective neck dissection and elective neck irradiation) with appropriate citations. 

Meta-analysis of 16 studies estimated an overall prevalence of occult metastases of 17%; this information is relevant and it could help clinicians in the decision making process. 

A considerable limitation of this meta-analysis was the absence of any subgroup analysis according with the site, staging and treatment, furthermore it was not possible to estimate the prognostic role of occult neck metastases. 

I advise the Authors to contact the Authors of the 16 studies included in the meta-analysis in order to potentially obtain more informations that could allow subgroup analysis.  

Author Response

We appreciate and agree with reviewer's comment. We acknowledge that this analysis presents a strong limitation, that is the impossibility to perform a subgroup analysis, preventing more interesting conclusions or considerations. This is highlited both in the results and discussion section. We tried to overcome this limit trough a descriptive report of the results of each paper selected, presenting dedicated sections in our discussion. We contacted all the authors in order to get more data about our search (we added a sentence about this in the methods section). Unluckily the paucity of data retrieved prevented any subgroup analysis. Results and limitations of our search should push future multicentre studies about this topic.

Round 2

Reviewer 1 Report

There was no appreciable change from the original manuscript with the exception of the addition of a single line about contacting the authors of the original manuscripts so my review comments are unchanged.